# Exploring the Interplay Between Healthcare Quality and Economic Viability Through Massive Data Analysis-Driven Multi-Hospital Management in a Spanish Private Multi-Hospital Network

**DOI:** 10.3390/healthcare13233034

**Published:** 2025-11-24

**Authors:** David Baulenas-Parellada, Javier Villalón-Coca, Daniel Vicente-Gallegos, Sandra Paniagua-Sánchez, Xavier Corbella-Viros, Angel Ayuso-Sacido

**Affiliations:** 1Dirección Corporativa Asistencial, Calidad, Innovación y Docencia, Grupo Hospitales Vithas, 28043 Madrid, Spain; dbaulenas@vithas.es (D.B.-P.); villaloncjf@vithas.es (J.V.-C.); 2Facultad de Medicina, Universidad CEU San Pablo, 28668 Madrid, Spain; 3Área de I+D+i, Fundación Vithas, Grupo Hospitales Vithas, 28043 Madrid, Spain; danivigallegos@gmail.com (D.V.-G.);; 4Departamento de Medicina, Facultat de Medicina i Ciències de la Salut, Universitat Internacional de Catalunya, 08036 Barcelona, Spain; xcorbella@uic.es; 5Faculty of Experimental Sciences and Faculty of Medicine, Universidad Francisco de Vitoria, 28223 Madrid, Spain

**Keywords:** health care indicators, Self-Organizing Maps, decision making, organizational management, health care economics and organizations

## Abstract

**Background**: Hospital management increasingly requires integrating quality and economic performance metrics to ensure efficiency and sustainability. However, evidence on how hospital key performance indicators (KPIs) relate to financial outcomes remains scarce, particularly in private healthcare systems. **Objective**: To examine the relationships between hospital KPIs and two financial metrics—Sales and EBITDA (Earnings Before Interest, Taxes, Depreciation, and Amortization)—in a Spanish private multi-hospital network. **Methods**: This retrospective, observational, multi-center study analyzed a final dataset of 47 standardized KPIs from 14 hospitals in the Vithas network. KPIs were examined using Self-Organizing Maps (SOM), an unsupervised neural network technique, to identify patterns and temporal dependencies with financial outcomes at contemporaneous, 3-month, and 6-month horizons. Robustness was evaluated through sensitivity analyses of model stability, data completeness, and clustering consistency. **Results**: The SOM analysis revealed six distinct clusters of KPIs, reflecting logical and interconnected behaviors. Sales and EBITDA were strongly associated with scheduled activity and space occupancy in the immediate term, while quality-related KPIs such as patient satisfaction and accessibility influenced financial outcomes at 3 and 6 months. These patterns suggest that selected KPIs can serve as predictive tools for financial performance. **Conclusions**: SOM proved effective for uncovering complex, nonlinear relationships between KPIs and financial metrics in hospital management. The study provides an operational framework linking standardized KPIs to financial outcomes in private hospitals, with implications for forecasting and strategic planning. Future research should incorporate additional KPIs, updated datasets, and SOM variants to validate and extend these findings across diverse healthcare systems.

## 1. Introduction

The hospital management model in Spain combines public and private systems to meet the population’s healthcare needs. The public system is primarily managed by the National Health System (SNS), which provides free care through tax-funded hospitals and health centers. The private system, on the other hand, includes hospitals and health centers funded by private insurance, direct patient payments, and public–private collaboration agreements [1,2].

In macroeconomic terms, healthcare expenditure in Spain represents around 10% of the national GDP, with roughly two thirds funded by the public sector and one third by private sources [3]. The country has more than 450 hospitals, of which a substantial proportion are privately owned, and private providers deliver a significant share of elective surgery and outpatient activity [3]. In parallel, private health insurance coverage has steadily increased over the last decade, reinforcing the role of large private hospital groups as key actors in the provision of healthcare services.

The trend toward dual coverage has grown significantly in recent years, reaching an average of 25.9% by 2023 [3]. Despite this substantial increase in activity, the private system has seen little change over the past 40 years, aside from the consolidation into large groups in both insurance and healthcare provision, neither the insurance model, the provision model, nor their relationship has evolved [4]. This relationship remains rooted in regulated fee-for-service payments, showing significant sustainability challenges such as lack of profitability, which can result in a decline in the quality of care, posing a particular risk to public–private partnerships and the balance between insurers and healthcare providers [2]. The need for new management models that integrate quality, safety, and patient experience is crucial to independently improve economic outcomes and ensure long-term sustainability.

Consistently, a positive correlation is observed between the financial health of hospitals and the quality of care they [5,6]. Hospitals with better financial performance tend to achieve better outcomes in terms of quality and patient safety [7]. Effective leadership, patient focus, process improvement and strong relationships with suppliers seems to be crucial for improving both financial performance and quality of care [8,9]. More specifically, quality measurement has a positive impact on the financial health of hospitals [10]. Patient care improvement also contributes positively [7]. Additionally, value-based quality assessment supports better financial outcomes [11]. However, the current evidence on the connection between quality indicators and economic outcomes, as well as the strength of this relationship, remains insufficient [12]. This highlights the importance of developing comprehensive analyses that examine both care-related and economic indicators to bridge this gap and support more informed and effective management strategies.

Vithas is a privately owned healthcare company founded 14 years ago, consisting of 21 hospitals and 36 medical centers distributed throughout the country, serving more than 5.5 million visits annually. Nonetheless, the hospital displays high heterogeneity in terms of case mix, complexity, size, and reference population, which necessitates the standardization of a minimum set of data for effective management. The unification of the various information systems across the 21 hospitals has enabled us to access a large volume of homogeneous and anonymized data to explore associations between different quality indicators and economic outcomes.

However, selecting the most suitable indicators to evaluate the performance of the various units within the group can be a challenge. Several authors have recently attempted to address this issue, related to timeliness, patient-centeredness, efficiency, utilization, clinical effectiveness, safety, responsive governance, staff orientation, and equity [13].

A cornerstone in this field is the framework proposed by Donabedian, who conceptualized healthcare quality into structure, process, and outcomes [14,15]. Building on this tradition, the Institute of Medicine (IOM), (now National Academy of Medicine), defined six essential aims for healthcare improvement: safe, effective, patient-centered, timely, efficient, and equitable [16]. These six aims were later operationalized and popularized under the acronym STEEEP to guide quality measurement in hospital organizations [17]. Together, these approaches emphasize the complexity of evaluating hospital performance and the need to consider multiple dimensions.

Additionally, Artificial Intelligence (AI) is transforming hospital management by enhancing efficiency, optimizing resources, and revolutionizing patient care [18]. The use of AI-based algorithms is also crucial for exploring associations between healthcare management indicators. These advanced tools can analyze large volumes of data and identify patterns that would be difficult to detect otherwise [19,20,21]. However, there are no standardized models for this purpose, which limits consistency and comparability in findings.

The motivation for this study stems from the limited evidence linking standardized hospital management indicators with financial performance in private multi-hospital networks. Although quality, accessibility, and efficiency metrics are widely used in routine monitoring, few studies have examined how these indicators interact with economic outcomes in real-world operational settings. This gap constrains data-driven decision-making, impedes the prioritization of investments, and limits understanding of how improvements in care delivery translate into financial sustainability. Addressing this need requires analytical frameworks capable of integrating diverse hospital KPIs and exploring their contemporaneous and lagged associations with economic performance.

In this study, we aim to identify the relationships between key hospital management indicators and two main business indicators: sales and EBITDA. (Earnings Before Interest, Taxes, Depreciation, and Amortization) by using AI based analytical models. Following a thorough evaluation of indicators and AI-based assessment models, we identified 58 key indicators and employed a Self-Organizing Map (SOM) model to uncover relationships among them. The SOM analysis revealed logical patterns and comparable behaviors among the indicators, which could be instrumental in forecasting financial performance in future periods.

To guide the study, we developed a conceptual model linking quality frameworks, operational indicators, analytical approaches, and financial outcomes (Figure 1). Building on the STEEEP domains—safe, timely, effective, efficient, and patient-centered—we operationalized hospital performance into a standardized set of management indicators. These indicators were then processed with advanced data-driven analytical techniques to uncover multivariate structures and patterns. Finally, the resulting indicator groups were connected to financial outcomes, measured through Sales and EBITDA at contemporaneous and lagged time horizons. This framework provides the theoretical and operational foundation of our analysis.

Despite the increasing attention to the relationship between healthcare quality and financial performance, most prior studies have focused on public healthcare systems or single clinical domains, often relying on linear statistical methods. Evidence from private hospital networks in Spain is scarce, and advanced analytical approaches capable of capturing non-linear interactions remain underexplored. This study addresses this gap by applying a Self-Organizing Map (SOM) framework to a standardized set of management indicators within the Vithas hospital group.

The specific objectives are: (i) to identify the hospital management indicators most strongly associated with financial outcomes (Sales and EBITDA); (ii) to examine how these associations evolve at contemporaneous, short-term (3-month), and medium-term (6-month) horizons; and (iii) to evaluate the robustness and practical utility of Self-Organizing Maps (SOM) as a scalable tool to integrate quality and financial performance in hospital management.

## 2. Materials and Methods

To achieve our objective of identifying the relationships between key hospital management indicators and two dependent business indicators (sales and EBITDA)**,** we adopted a comprehensive methodological approach that integrated data collection, quality assurance, and advanced analytical techniques. Utilizing data from the Vithas Management System (VMS), we standardized and consolidated operational and clinical information across the hospitals. We then employed artificial intelligence (AI) based analytical models, specifically Self-Organizing Maps (SOMs), to uncover complex patterns and associations among the selected indicators. This approach enabled us to effectively explore the potential predictive relationships between hospital management practices and financial performance, aligning with our goal of enhancing both quality of care and economic outcomes.

### 2.1. Study Design and Research Questions

This study employed a retrospective, observational, multi-center design, analyzing routinely collected management and quality data from 16 hospitals of the Vithas Group over a three-year period (January 2019 to December 2021). The observational unit is the hospital-month. This period was chosen because it represents the most recent interval with complete and homogeneous data across the network, prior to the replacement of the corporate management system in 2022, ensuring robustness and comparability.

Data corresponds to aggregated operational and clinical indicators obtained from corporate information systems. To avoid bias caused by the anomalous activity patterns during the COVID-19 outbreak, the months of March, April, and May 2020 were excluded from the analysis.

The central research question was how hospital quality and operational indicators, when structured under the STEEEP framework, relate to financial outcomes (Sales and EBITDA) in Spanish private hospitals at contemporaneous, short-term (3-month), and medium-term (6-month). Our overall hypothesis was that different domains of hospital performance—activity, quality, and accessibility—would be associated with these financial outcomes over distinct time horizons.

Ethical considerations: The study was conducted exclusively with aggregated and anonymized hospital management indicators routinely collected at the institutional level. No patient-level or identifiable data were used at any stage. In accordance with the General Data Protection Regulation (GDPR) and national requirements, institutional review board approval and informed consent were not applicable.

### 2.2. KPI Framework and Operationalization

The definition of indicators was guided by the STEEEP framework (safe, timely, effective, efficient, and patient-centered), complemented by Donabedian’s structure–process–outcome model. In the context of Spanish private hospitals, the equity domain from the original STEEEP framework was excluded, as it is primarily related to access and financing mechanisms specific to the public system and therefore not measurable within this network. A multidisciplinary governance committee, composed of representatives from clinical operations, patient experience, quality and safety, coding and documentation, and finance, reviewed both corporate dashboards and the scientific literature to translate each STEEP domain into measurable indicators available in the Vithas Management System. This process yielded a set of corporate reference KPIs, which were subsequently organized into five operational domains. Safety included indicators such as compliance with surgical checklists and anesthesia or surgery consents. Timeliness was reflected in metrics like emergency room waiting times and time to first consultation. Effectiveness comprised complication ratios and mortality indicators. Efficiency was captured through operating room occupancy, average length of stay, bed occupancy, and outpatient-to-surgery ratios. Finally, patient-centeredness was represented by Net Promoter Scores across outpatient consultations, surgery, hospitalization, and emergency care. This STEEP-based KPI framework constituted the foundation for the subsequent analysis of their relationship with financial outcomes, yielding an initial set of 52 corporate reference KPIs.

All KPIs were expressed using standardized and homogeneous units across hospitals. Activity indicators were measured as absolute counts (e.g., number of surgeries, ER visits, outpatient consultations). Efficiency indicators were expressed as ratios or percentages (e.g., operating room occupancy %, bed occupancy %, outpatient-to-surgery ratio). Quality indicators were reported as percentages or standardized scores (e.g., Net Promoter Score 0–100). Full definitions, units of measurement, and calculation rules for each KPI are provided in Appendix A.

### 2.3. Data Sources and Standardization

The tools selected to obtain the information are the following: Benchmarking Sanitario 3.0 (BS3) (Barcelona, Spain, 2023): Used to compare performance within the network of facilities or against industry standards, which is essential for maintaining operational and competitive efficiencies. Basic Minimum Data Set (CMBD): In compliance with Royal Decree 69/2015 regulating the Registry of Specialized Care Activity (RAE-CMBD), the CMBD collects essential administrative and clinical data, facilitating quality monitoring and effective resource planning. Qlik Sense (Qlik, Berks County, PA, USA, 2023): A tool for advanced data visualization and interactive analysis. It enables the discovery of trends and insights across various data systems, which is crucial for maintaining operational efficiencies. Indaga Reclama and Indaga Analytics (in-house development, last updated 2023): Two tools focused on data exploration and analysis. Indaga Reclama is geared towards claims and complaint management, while Indaga Analytics provides broad data analysis capabilities. SuccessFactors (SAP, Walldorf, Germany, 2023): Used for advanced human resources management, integrating functions such as talent management, performance tracking, and employee development. It enhances operational efficiency by streamlining recruitment, providing real-time feedback, and facilitating e-learning tailored to individual career paths. Finally, the Clinical History Review ensures compliance with health regulations and enhances patient care quality by providing a comprehensive view of clinical histories.

To ensure standardization across centers, all indicators were defined with homogeneous denominators, units of measurement, and calculation procedures, aligned with corporate governance. Data from the different sources were consolidated daily into a central Data Warehouse, which guaranteed that the dataset was consistent, comparable, and traceable to source systems.

### 2.4. Data Preparation and Quality Control

Prior to analysis, several quality-control procedures were applied to ensure the consistency and reliability of the dataset. Indicators with more than 10% missing values were excluded, as proportions above this threshold are known to threaten the validity of health research [22]. For indicators with minor gaps (≤10%), values were imputed using the nearest available temporal observation within the same hospital. This conservative approach, commonly applied in longitudinal health services datasets, preserves continuity, avoids the introduction of artificial trends, and ensures comparability across centers.

However, all KPIs were normalized using min–max scaling (range 0–1). This procedure was applied to ensure comparability among indicators with different measurement units and scales (e.g., percentages, ratios, or absolute counts). Normalization prevents indicators with large numeric ranges from dominating the SOM learning process and allows each KPI to contribute equally to the map topology. This approach is particularly appropriate in our dataset, where most KPIs are percentages, ratios, or standardized indicators that already adjust for hospital size, thus minimizing the risk of scale-related distortions.

Outliers were defined as observations exceeding three standard deviations (SD) from the mean, calculated separately for each hospital and KPI. Although this rule originates from the properties of the normal distribution [22], no strict normality was assumed, as the dataset involved many KPIs, several hospitals, and relatively few monthly observations per series. Under these conditions, formal tests of normality were neither feasible nor reliable. Instead, the ±3 SD threshold was applied as a conservative empirical rule, widely used in health services research to detect extreme institutional performance values [23], and adopted here as a pragmatic quality-control criterion

To avoid redundancy and potential multicollinearity, pairwise correlations were calculated across the initial set of indicators. A threshold of |r| > 0.90 was applied, consistent with health services research and multivariate analyses [24]. The full correlation matrix is provided in Appendix A.

Hospital homogeneity was assessed using hierarchical clustering (Ward’s method, Euclidean distance), which allowed us to evaluate whether all centers presented comparable profiles. The resulting dendrogram is provided in Appendix A. Hospitals with markedly distinct patterns were excluded from the dataset to avoid bias in the identification of global performance patterns.

### 2.5. Final Dataset Composition

After applying the predefined quality-control procedures (correlation filtering, outlier detection, missing-data imputation, and hospital homogeneity analysis), the initial set of 52 KPIs was reduced to a final set of 47 KPIs from 14 hospitals, yielding a total of 21,714 monthly observations collected between January 2019 and December 2021. Specifically, three KPIs were excluded due to redundancy (|r| > 0.90, see Appendix A), one KPI due to excessive outliers (see Appendix A), and one KPI due to excessive missing data. In addition, two hospitals (Vithas Lleida and Vithas Xanit) were removed after the homogeneity analysis. To avoid bias caused by the anomalous patterns during the COVID-19 outbreak, the months of March, April, and May 2020 were excluded.

The detailed list of the 47 KPIs finally included, together with their abbreviations, is provided in Appendix A, which also reports descriptive statistics (mean, median, SD, quartiles, kurtosis) for transparency in the dataset preparation

### 2.6. Analytical Approach (Overview)

The analytical strategy was designed to identify patterns among hospital KPIs and to examine their relationships with financial outcomes. Sales and EBITDA were analyzed contemporaneously and with 3- and 6-month lags to explore potential temporal dependencies. Different artificial intelligence models commonly used in health services research were considered as potential analytical tools.

#### 2.6.1. Rationale for Selecting SOM

Five artificial intelligence models commonly used for analyzing large datasets were considered—Support Vector Machines (SVM), Linear Discriminant Analysis (LDA), Random Forest (RF), k-Nearest Neighbors (k-NN), and Self-Organizing Maps (SOM). Their suitability was assessed qualitatively according to four predefined criteria: accuracy, scalability, interpretability, and computational efficiency [25]. The assessment was performed by an AI expert on the team and supported by methodological evidence from the literature [26,27,28,29,30,31,32].

While each method presented specific strengths and limitations, SOM stood out as the most balanced option for the objectives of this study. In particular, SOM integrates the capacity to capture non-linear relationships with the advantage of producing intuitive two-dimensional visualizations, making it especially suited for hospital management contexts where interpretability is essential. A detailed comparative summary is provided in Appendix A.

#### 2.6.2. SOM Configuration

For the implementation of the SOM, six indicators corresponding to the dependent financial metrics (Sales and EBITDA, contemporaneous and lagged) were excluded from the input space, so the map was trained using the remaining 41 KPIs. These dependent variables were not part of the training phase but were subsequently projected onto the trained SOM to examine their relative positioning with respect to the clusters of management indicators.

To prevent any form of data leakage, SOM was trained exclusively with contemporaneous KPI values. Sales and EBITDA at 0, 3, and 6 months were not included in the training dataset and were projected only after the map was fully trained. This guarantees that the SOM topology was determined solely by KPI patterns. Figure 2 shows the analytical workflow of the study.

A hexagonal grid of 9 rows × 12 columns was defined using the function somgrid. The hexagonal topology was selected because it preserves neighborhood relations more effectively than rectangular grids, allowing smoother data representation and continuity between nodes. The grid size was determined according to established heuristics that balance sample size and dimensionality, ensuring sufficient resolution while avoiding overfitting [26].

Training was performed with iterative updates and a decaying learning rate, starting at 0.05 and decreasing linearly to 0.01. This schedule follows standard recommendations, beginning with a relatively high rate to allow rapid global ordering of neurons, followed by progressive refinement with smaller values. A Gaussian neighborhood function was applied, with an initial radius of 2.65 that decreased progressively, ensuring a smooth transition from global to local structure refinement [26].

The SOM was trained for 100 iterations, which proved sufficient to stabilize grid organization. This choice is consistent with previous methodological work indicating that 100–200 iterations are typically adequate for medium-sized datasets in healthcare and biomedical applications [33]. No additional stopping criterion was applied. Euclidean distance was used as the similarity metric.

To support reproducibility, the random seed was fixed, following best practices in computational research reproducibility.

#### 2.6.3. Implementation Details in R

All analyses were conducted in R software, version 4.2.2 (R Foundation for Statistical Computing, Vienna, Austria). General-purpose packages such as *dplyr*, *ggplot2*, *readxl*, *xlsx*, *ISLR*, and *MASS* were used for data manipulation, statistical operations, and visualization. For the implementation of Self-Organizing Maps, the *kohonen* and *aweSOM* packages were employed, providing the functions required to train, execute, and represent the maps. Supplementary packages such as *htmlwidgets* supported interactive display of results. Together, these tools ensured reproducibility of the full analytical workflow [26].

### 2.7. Robustness and Validation Tests

The robustness and validity of the SOM were assessed through three complementary procedures.

First, sensitivity to initialization and parameter settings was examined. The SOM was trained with multiple random seeds and alternative network configurations, including variations in several arguments. Although minor differences in cluster compactness were observed, the overall map topology and cluster structure remained consistent across repetitions (Appendix A).

Second, robustness against partial data exclusion was tested by randomly removing 20% of the hospitals and, in a separate test, by excluding the last 3 months of data before retraining the models. In both cases, the resulting maps preserved the general organization and relative positioning of indicators, confirming that the observed patterns did not depend on the inclusion of all centers or periods (Appendix A).

Third, as a benchmark, a conventional k-means clustering was applied to the same dataset. The partition was less informative than the SOM because k-means ignores data topology and only groups points by distance to centroids. In contrast, SOM preserved neighborhood relationships, providing a richer representation of indicator interdependencies (Appendix A).

Together, these tests demonstrate that the SOM configuration applied in this study is stable, reliable, and resilient to variations in initialization, parameter settings, and data completeness. To support reproducibility, Euclidean distance was consistently used as the similarity metric after min–max normalization of all KPIs, and the random seed was fixed following best practices in computational research reproducibility.

## 3. Results

Analyses were conducted on the standardized dataset of 47 key performance indicators (KPIs) from 14 hospitals (21,714 monthly observations; see Methods, Section 2.5 and Appendix A). The application of Self-Organizing Maps (SOM) revealed significant relationships between these KPIs and the two main economic performance metrics (Sales and EBITDA). In addition, temporal dependencies were identified at contemporaneous, 3-month, and 6-month horizons.

The results are organized into two sections. First, we present the SOM analysis of hospital management KPIs, which identified clusters of indicators with similar behavior patterns (Section 3.1). Second, we examine the positioning of the two dependent financial metrics—Sales and EBITDA—within the SOM at contemporaneous, 3-month, and 6-month horizons (Section 3.2).

### 3.1. Hospital Management KPIs Analyzed by SOM

The SOM was trained using the 41 KPIs derived from the STEEP framework. To facilitate interpretation, KPIs located in the same node or in adjacent nodes were grouped, as they share similar behavior patterns. Although clustering was not an explicit study objective, the SOM methodology logically organized the KPIs into six clusters. This emergent classification facilitated the interpretation of interrelated behaviors and their subsequent associations with financial outcomes, and proved consistent across different periods and hospitals (Figure 3).

Group A comprised KPIs of scheduled activity and occupancy, including operating room occupancy, available OR hours, bed occupancy, complexity ratio, first medical and surgical visits, high-complexity procedures, surgical discharges, total births, and total surgeries. Group B included process quality KPIs and inpatient satisfaction, specifically the completeness of discharge reports and medication orders, together with the NPS for hospitalization. Group C consolidated KPIs of surgical quality and outpatient satisfaction, such as compliance with surgical checklists, NPS for outpatient consultations and outpatient surgery, the outpatient surgery ratio, and the inverse readmission ratio. Group D covered urgent activity, represented by adult and pediatric emergency visits and medical hospital discharges. Group E aggregated accessibility and continuity-of-care indicators, including the proportion of ER patients referred to outpatient services, ER timeliness within 30 min, the proportion of triaged ER patients, the global NPS, and the inverse mortality ratio. Finally, Group F mainly reflected documentation quality and clinical evaluation, with KPIs such as compliance with anesthesia consents, early discharges before noon, completeness of nursing evaluations, completeness of surgical reports, the diagnosis intensity ratio, and the inverse waiting time for first consultations.

A small set of KPIs did not cluster consistently (non-grouped, NG), including the percentage of urgent surgeries, the number of new patients in consultations, and the NPS for emergency services.

### 3.2. Analysis of the Dependent Indicators: Sales and EBITDA

After an initial analysis of the SOM network structure, the two financial KPIs (Sales and EBITDA) were subsequently projected onto the trained map to assess their positioning relative to the identified groups. To explore potential temporal dependencies, both indicators were also analyzed with 3- and 6-month lags, reflecting short- and medium-term effects frequently observed in hospital management cycles (Figure 4).

At the contemporaneous horizon, Sales and EBITDA were both located within Group A, which predominantly included indicators of scheduled activity volume and space occupancy, such as number of surgeries performed, available operating room hours, and hospital discharge rates. This placement highlights the immediate dependence of financial performance on activity and occupancy levels.

When assessed with a 3-month lag, Sales shifted to Group C, characterized by surgical process quality and patient satisfaction in scheduled procedures, while EBITDA moved to Group E, which encompassed accessibility and continuity-of-care indicators in emergency services (e.g., patient satisfaction in ER and timeliness of urgent care). These shifts indicate that improvements in surgical quality and patient experience exert financial impact within three months, whereas accessibility and emergency performance influence profitability over a similar timeframe.

At the 6-month lag, Sales remained in Group C but migrated closer to the map center, suggesting broader interactions with multiple KPI domains. Conversely, EBITDA moved away from Group E and toward the SOM center, without a clear grouping, implying that medium-term profitability becomes influenced by a more diffuse set of factors beyond emergency accessibility alone.

## 4. Discussion

### 4.1. Findings

Our data revealed several critical insights into the relationships between hospital quality indicators and economic outcomes. Specifically, the analysis using Self-Organizing Maps (SOM) demonstrated a logical clustering of indicators into six distinct groups, each reflecting interconnected behavior patterns. These clusters highlighted associations such as the alignment of scheduled activity volume and space occupancy indicators with immediate financial outcomes like sales and EBITDA, and the delayed impact of quality-related metrics on financial performance over three- and six-month periods.

The methodology used can add value to the current decision-making processes in healthcare organizations. However, it is important to emphasize the need for a well-reasoned selection of indicators to achieve useful results, while also being aware that the STEEEP methodology may have various adaptations depending on the financing model, among other criteria [12,13,18,34]. Expanding the selection of indicators is crucial, with a focus on strengthening underrepresented areas such as human resource management, which plays a significant role in influencing care delivery. Additionally, it is also important to emphasize the need for complete and high-quality information to accurately assess hospital performance and its connection to economic outcomes. Reliable and comprehensive data derived from these indicators allow for the identification of significant patterns, such as temporal dependencies and relationships between operational metrics and financial performance, which are critical for informed decision-making and strategic planning in healthcare management.

The analysis of hospital clustering revealed both homogeneity and variability in the behavior of the selected indicators. Among the fourteen hospitals included, three distinct groups emerged: two with five hospitals each and one with the remaining four. These groupings appear logical when external factors, such as geographic location (e.g., being in a major city or near a beach), are considered, as they likely to influence operational and patient demographic metrics. The observed clustering underscores the interplay between inherent hospital characteristics and their performance indicators, highlighting the importance of considering contextual factors when interpreting and comparing hospital data. This variability within clusters suggests that while certain indicators exhibit consistency across similar environments, other metrics may vary significantly, necessitating tailored management approaches to address unique challenges and optimize performance within each group.

### 4.2. Implications

The analysis of the dependent indicators, Sales and EBITDA, provides critical insights into their dynamic relationships with hospital performance metrics. By integrating these financial indicators into the SOM framework, both with and without temporal lags, the study uncovers significant connections between operational activities, quality indicators, and financial outcomes. From the perspective of hospital operations analysis, the placement observed is consistent, as sales and EBITDA are intrinsically linked to the volume of hospital activity. A higher number of scheduled services typically results in increased revenue and enhanced financial performance for the organization.

The observed shift in billing to Group C after a 3-month lag suggests that improvements in the quality of surgical processes and patient satisfaction in scheduled processes directly influence financial performance within a relatively short period. This highlights the importance of investing in quality enhancement initiatives, as they can lead to tangible financial benefits within three months.

Similarly, the movement of profit margin to Group E suggests that enhancements in accessibility and continuity of care in emergency services have a positive impact on financial performance, but this effect becomes apparent after three months. This delay underscores the need for sustained efforts in improving emergency care services to achieve long-term financial sustainability.

From an operational analysis perspective, the persistence of sales within Group C highlights a sustained relationship between the quality of surgical processes, patient satisfaction in scheduled services, and revenue generation. The movement of sales closer to the center of the SOM suggests an increasing interaction with a broader set of KPIs, indicating more complex and challenging-to-identify influences on sales performance over time. The relocation of EBITDA from Group E to the center of the SOM can indicate departure from its association with emergency services accessibility and continuity of care. This shift may suggest that, over a longer period, EBITDA becomes influenced by a wider array of factors beyond emergency service performance. This movement towards the center of the SOM could indicate that EBITDA is beginning to exhibit patterns that are less aligned with the used STEEP KPIs, potentially making its behavior more challenging to predict. The displacement of Sales and EBITDA across the 0-, 3-, and 6-month projections should be interpreted as a directional tendency within the KPI space rather than as a statistically tested effect. As SOM is an unsupervised, distance-based model, these movements do not correspond to parametric significance testing but reflect geometric consistency. The fact that both indicators shift toward nodes associated with better operational and patient-experience performance suggests a meaningful pattern rather than random variation.

Integrating financial indicators with temporal lags into the SOM analysis provides valuable insights into how quality metrics influence financial performance over time. The identified relationships underscore the critical role of quality management in achieving both clinical excellence and economic sustainability. Future research should explore these temporal dynamics further and consider additional quality KPIs to build a more comprehensive model of hospital performance.

### 4.3. Comparison with Previous Literature

Our findings lie at the intersection between classical quality assessment frameworks and unsupervised learning methods. On the one hand, Donabedian’s structure–process–outcome model (1966, 1988) and the six quality aims defined by the Institute of Medicine (2001) [16], later operationalized as the STEEEP framework [17], established the foundation for structuring hospital performance measurement. On the other hand, the Balanced Scorecard applied in healthcare highlighted the need to align clinical and financial dimensions, although with tensions in its practical implementation [21]. The choice of Self-Organizing Maps (SOM) as a tool to uncover multivariate patterns builds on previous experiences demonstrating its usefulness and interpretability in specific contexts such as hemodialysis clinics [20]. In contrast, systematic reviews suggesting quality–finance links emphasize methodological heterogeneity [5], while longitudinal studies in focused areas such as cardiac surgery show associations with future financial outcomes [10]. Our study contributes to this line of research by providing a homogeneous multicenter dataset and an analytical architecture able to distinguish contemporaneous and lagged effects (0, 3, and 6 months). Table 1 summarizes these references and highlights the differential contribution of our analysis in a Spanish private hospital network.

Table 1 Comparative summary of key studies linking hospital quality and financial performance. Our study advances this literature by applying SOM to standardized STEEP-based KPIs in a Spanish private hospital network, directly connecting quality domains with Sales and EBITDA.

Taken together, prior literature legitimizes both quality–management frameworks and the use of SOM to reveal latent patterns. Our contribution lies in integrating these approaches and linking them to Sales and EBITDA at different time horizons, under standardized corporate definitions and with robustness checks. This approach offers hospital management an operational perspective on which domains to prioritize and when to expect financial impact.

### 4.4. Policy Implications

The results of this study have relevant implications for hospital management and policy-making within private multi-hospital networks. The observed temporal relationships between quality, accessibility, and financial outcomes suggest that management teams should incorporate KPI-based monitoring systems to identify early signals of operational and economic risk. Prioritizing interventions in domains such as scheduled activity, patient experience, and emergency accessibility may help improve both clinical performance and profitability. In addition, integrating STEEEP-based indicators into financial planning cycles could support more balanced decision-making, enhance transparency in public–private collaboration models, and facilitate benchmarking across hospitals. These insights provide a foundation for developing evidence-based policies that align operational performance with long-term economic sustainability.

## 5. Conclusions

The study identified 47 key performance indicators (KPIs) for hospital management and applied Self-Organizing Maps (SOM) to analyze their interrelationships. The SOM methodology proved effective in organizing these indicators into coherent structures and revealed significant associations with financial outcomes such as Sales and EBITDA, including temporal dependencies that suggest potential predictive value. Building on the STEEEP methodology, the analysis operationalized classical quality domains—safety, timeliness, effectiveness, efficiency, and patient-centeredness—and linked them to economic sustainability.

These findings reinforce the utility of advanced data analysis techniques in hospital management, as they allow the detection of complex relationships that traditional methods often overlook. They also highlight the potential of integrating operational and quality metrics with financial indicators to optimize decision-making and strengthen the viability of healthcare organizations. The study provides a scalable and adaptable framework that can be expanded with additional indicators and methodological refinements in future work. Overall, it demonstrates the relevance of comprehensive and reliable data, together with an integrated quality framework, to guide hospital management and strategic planning.

Nevertheless, these results should be interpreted with caution, as they are based on retrospective data from a Spanish private multi-hospital network and may not be directly generalizable to public healthcare systems or other countries without additional validation. Future work should incorporate updated datasets, additional KPIs—such as those related to human resources—and comparative analyses using alternative SOM variants or clustering methods to confirm the robustness of these findings.

Overall, this study provides a scalable and adaptable framework that demonstrates the relevance of comprehensive and reliable data, together with an integrated quality framework, to guide hospital management and strategic planning.

## 6. Limitations

Beyond these considerations, some methodological limitations must also be acknowledged. As the analysis was performed at an aggregated hospital-month level, all associations identified are ecological in nature and cannot be interpreted at the patient or individual process level. The relationships observed should therefore be understood as hospital-level patterns rather than causal effects on specific clinical pathways or patient subgroups. Self-Organizing Maps (SOMs), like other analytical methods, have inherent constraints, such as their rigid and fixed neighborhood topology, which cannot be modified during the learning phase. Nevertheless, SOMs proved effective in this context for extracting insights from the KPI framework employed by the Vithas Group and for uncovering complex relationships that traditional techniques would likely overlook [26,27,32].

Regarding the study period, the dataset (2019–2021) represents the most recent interval with complete and homogeneous data across the Vithas network before the corporate management system was replaced in 2022. Although the COVID-19 pandemic introduced short-term fluctuations in activity and case-mix, the corporate definitions and calculation rules for all KPIs remained unchanged, allowing comparability across the period. While this ensured robustness and comparability, newer data generated under the updated system should be incorporated in future research to validate the stability of the observed patterns. In addition, although missing values were relatively infrequent and imputed using conservative temporal methods, this process may still introduce minor bias. Finally, two hospitals were excluded due to heterogeneous performance profiles, which may limit the generalizability of the findings.

Future research should explore methodological enhancements to improve the analytical capabilities of SOM. These include testing alternative distance measures (e.g., Manhattan, Mahalanobis) beyond Euclidean distance, developing individualized maps, and conducting comparative analyses with other clustering techniques. In addition, established SOM variants such as the Growing SOM (GSOM), Growing Hierarchical SOM (GHSOM), Fuzzy SOM, and Temporal SOM (TSOM) could be applied to healthcare management datasets to provide greater flexibility and adaptability over time.

## Figures and Tables

**Figure 1 healthcare-13-03034-f001:**
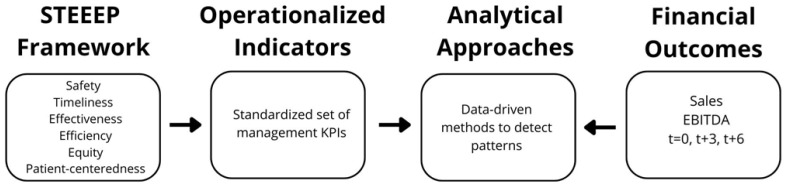
Conceptual model. The STEEEP framework was translated into a standardized set of hospital management indicators. These indicators were analyzed using advanced data-driven methods to uncover multivariate patterns. The resulting clusters were then related to financial outcomes (Sales and EBITDA) measured contemporaneously and at lagged intervals.

**Figure 2 healthcare-13-03034-f002:**
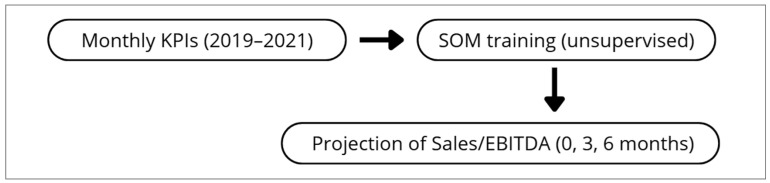
Analytical workflow. Monthly KPIs (2019–2021) were used to train the SOM. Sales and EBITDA at 0, 3, and 6 months were projected only after training.

**Figure 3 healthcare-13-03034-f003:**
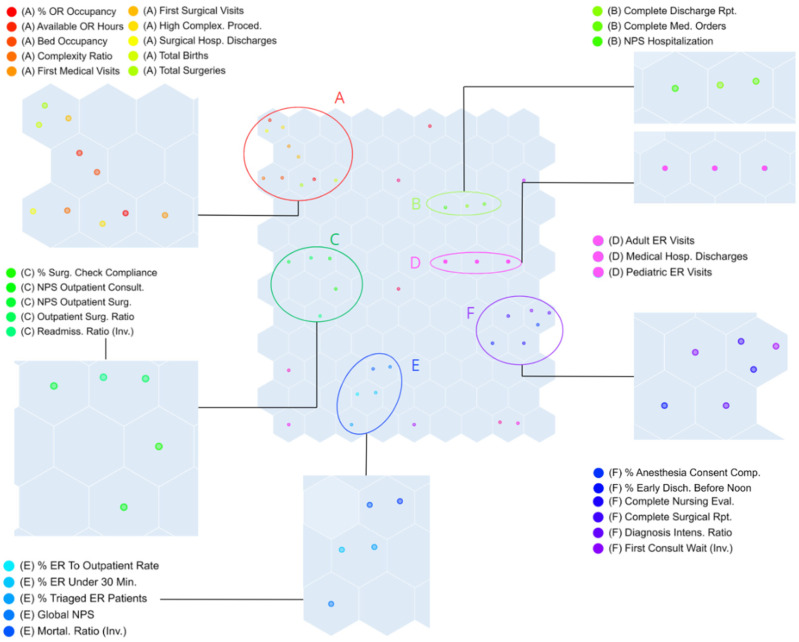
Self-Organizing Map displaying the location of STEEP-based KPIs according to their behavior patterns. Six clusters (A–F) are observed, defined by groups of KPIs located in the same or adjacent nodes. Each hexagon represents a SOM node grouping similar hospital metrics, with proximity indicating greater similarity and distance reflecting different patterns. Colors and labels (A–F) identify the six clusters, while expanded panels show the variables included in each group.

**Figure 4 healthcare-13-03034-f004:**
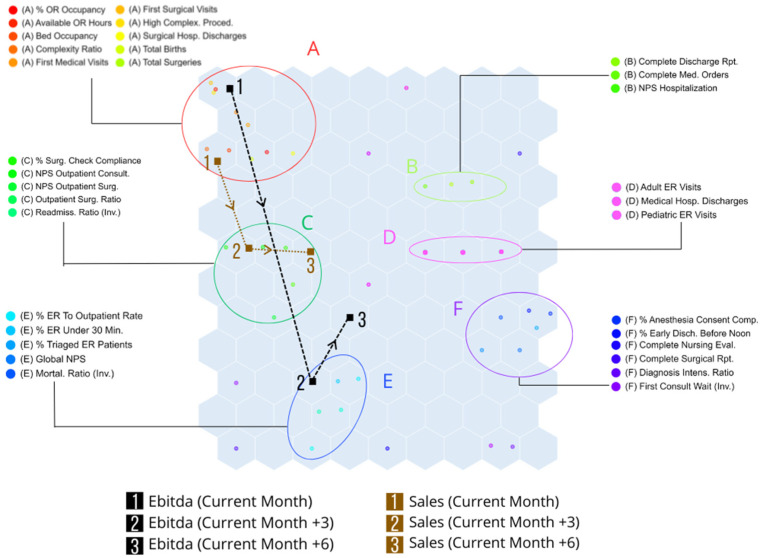
SOM including the two financial indicators, Sales and EBITDA, analyzed contemporaneously and with 3- and 6-month lags. The same clusters (A–F) identified in Figure 4 are maintained to illustrate temporal relationships between quality KPIs (independent variables) and financial outcomes (dependent variables). Sales are represented by brown squares and EBITDA by black squares, with arrows indicating their positional changes across time horizons.

**Table 1 healthcare-13-03034-t001:** Comparative summary of previous literature on hospital quality and financial performance.

Author/Year	Context/Scope	Methodology	Main Findings	Contribution of Our Study
Donabedian, 1966; 1988 [14,15]	Conceptual framework for healthcare quality	Theoretical (structure–process–outcome)	Defined the classical triad (structure, process, outcomes) for quality assessment.	Conceptual foundation for structuring hospital performance evaluation.
Institute of Medicine, 2001 [16]	US healthcare system reform	Policy report (Crossing the Quality Chasm)	Defined six aims: safe, effective, patient-centered, timely, efficient, equitable.	Basis later operationalized as the STEEEP framework in hospital management [15].
Gurd & Gao, 2008 [21]	Balanced Scorecard in healthcare organizations	Review of BSC cases	Reviewed healthcare BSC applications; highlighted tensions between financial and quality dimensions.	We extend BSC logic by integrating STEEEP-based KPIs with SOM to capture non-linear patterns.
Cattinelli et al., 2012 [32]	Hemodialysis clinics (Italy)	SOM applied to BSC performance data	SOM identified patterns supporting BSC-based performance monitoring.	We broaden SOM to a multi-hospital network and link clusters with Sales and EBITDA.
Barnes et al., 2017/2018 [5]	US hospitals (systematic review)	Review of 13 empirical studies	Early evidence of a quality–finance link, with heterogeneity and limited consistency.	We contribute homogeneous, standardized multicenter data and lagged analyses.
Enumah et al., 2022 [10]	US hospitals with cardiac surgery	Longitudinal statistical analysis	Measured quality predicted improved future financial performance.	Supports lagged effects, which we operationalize at 0/3/6 months across STEEEP domains.
Baulenas et al., 2025. This study	Spanish private hospital network	SOM on 47 STEEP-based KPIs	Six KPI clusters with immediate and lagged financial impacts.	First to integrate Donabedian → IOM (six aims) → STEEP → SOM → financial outcomes in a private multi-hospital network.

## Data Availability

The original contributions presented in this study are included in the article/Appendix A. Further inquiries can be directed to the corresponding author.

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
