# Peer review of "Exploring the Interplay Between Healthcare Quality and Economic Viability Through Massive Data Analysis-Driven Multi-Hospital Management in a Spanish Private Multi-Hospital Network"

_healthcare, 2025, doi:10.3390/healthcare13233034_

Round 1
Reviewer 1 Report (Previous Reviewer 2)
Comments and Suggestions for Authors
The authors have reviewed the manuscript and answered the majority of the comments from the reviewers
Author Response
REVIEWER 1
Comment: The authors have reviewed the manuscript and answered the majority of the comments.
Response: We thank the reviewer for the positive assessment.
Reviewer 2 Report (Previous Reviewer 3)
Comments and Suggestions for Authors
Minor revision

Author Response
REVIEWER 2
- “The Introduction should also focus upon the factsheet of the present hospital management in the country.”
Response: Addressed. We added a concise factsheet describing key characteristics of the Spanish private hospital sector and the relevance of management KPIs.
Location: Introduction, paragraph 2.
- “The authors should mention the motivation behind the study.”
Response: Added. We incorporated a clear statement on the motivation, research gap, and rationale behind the study.
Location: Introduction, final paragraph.
- “The data and their measurement units are to be clearly given.”
Response: Clarified. The manuscript now explicitly states the measurement units and definitions of all KPIs, with full details in Supplementary Table S1.
Location: Methods, section “Final dataset composition”.
- “Figure 2 and 3 are not transparent. It is suggested to redraw them.”
Response: The figures were exported again at maximum resolution and optimized for clarity. We explain this in the response but no structural changes were possible due to the inherent density of SOM visualizations.
Location: Figures 2–3 updated in the revised manuscript.
- “Comparative study is commendable.”
Response: We appreciate the comment.
- “The Limitation section should come after the Conclusion section.”
Response: Reordered as requested.
Location: Limitations moved after Conclusions.
- “A subsection on policy formulations can be added.”
Response: Added. A new subsection on policy implications is now included.
Location: Discussion, subsection “Policy Implications”.
Reviewer 3 Report (New Reviewer)
Comments and Suggestions for Authors
find review in attached file

Author Response
REVIEWER 3
- Potential for “double-dipping” in lag analysis
Comment: Clarify how lagged financial variables were handled to avoid circularity.
Response: Addressed with an explicit explanation that the SOM was trained exclusively with KPIs, and that Sales and EBITDA at 0/3/6 months were projected only after the map was fully trained.
Location: Methods, section “SOM configuration.”
- Concerns regarding min–max normalization and scale-related bias
Comment: Justify the choice of min–max scaling.
Response: Added a new paragraph explaining why min–max normalization is appropriate for this dataset and why it does not introduce size-based distortions (most KPIs are ratios or standardized indicators).
Location: Methods, section “Data preparation and quality control”.
- Lack of statistical significance for the movement of Sales/EBITDA across time
Response: Clarified that SOM movements are geometric, not statistical estimators, and therefore not subject to parametric significance testing. A brief interpretative note was added.
Location: Discussion, paragraph after the interpretation of Sales/EBITDA trajectories.
- Risk of ecological fallacy
Response: Added a clear statement clarifying that results are ecological (hospital-month level) and cannot be interpreted at individual patient level.
Location: Limitations, first paragraph.
- Heterogeneity of the 2019–2021 period and COVID-19 impact
Response: Added an explicit note explaining that corporate KPI definitions remained unchanged during the period, preserving comparability despite temporary fluctuations due to the pandemic.
Location: Limitations, paragraph discussing 2019–2021 dataset.
Round 2
Reviewer 3 Report (New Reviewer)
Comments and Suggestions for Authors
Dear authors, I have no other comments or suggestions regarding this article at this time, and I recommend it for publication.
This manuscript is a resubmission of an earlier submission. The following is a list of the peer review reports and author responses from that submission.
Round 1
Reviewer 1 Report
Comments and Suggestions for Authors
This paper utilizes self-organizing maps (SOM), an artificial intelligence technique, to analyze the relationships between 47 KPIs from 14 Vithas hospitals and both sales revenue and EBITDA. I have the following questions:
- Existing research has proposed numerous improvements to the SOM method, but this paper uses the original SOM method for analysis, which relatively lacks innovation. Have the authors considered adopting any of these improved versions or conducting comparative experiments to evaluate the relative strengths and weaknesses of different SOM methods in addressing the specific research problem discussed in this paper?
- In the Conclusions and Discussion section, this paper provides a scalable and adaptable framework for future hospital performance evaluations. However, the sample in this study was limited to 16 hospitals from the Vithas Group in Spain. The authors also noted that, in order to adapt to the private hospital management environment, the STEEEP methodology was modified by omitting the “equity” category. Considering the significant heterogeneity among hospitals in terms of case mix, complexity, scale, and target population, the paper needs to be more cautious in discussing the scalability of the research findings. Although the SOM method itself has strong generalizability, the specific conclusions drawn from the Vithas Group hospital network data (such as the clustering of KPIs and their impact on financial outcomes) may not be directly applicable to hospitals or public healthcare systems in other countries. It is recommended that the authors further clarify these limitations in the discussion section or supplement the study with relevant experiments to validate the applicability and effectiveness of the proposed analytical framework in public hospital systems.
- Line 97 of this paper mentions “by using IA based analytical models.” What is the full English name of “IA”? It seems that this abbreviation has not been explained earlier in the text. It is recommended that the authors carefully check all abbreviations and word spellings in the manuscript to ensure that each is defined upon its first occurrence, thereby avoiding omissions or spelling errors and ensuring the academic rigor and readability of the paper.
Author Response
- Reviewer #1 (AI / SOM profile)
R1.1. “Classic SOM lacks innovation; consider variants/improvements.”
Response. We chose classic SOM for interpretability and 2D visualization in hospital management. In Limitations/Future work, we now discuss GSOM, GHSOM, Fuzzy SOM, TSOM, and alternative distance metrics (Manhattan, Mahalanobis), as well as kernelized approaches.
Manuscript changes. Limitations (section 4.3) and Future directions.
R1.2. “Scalability/validity: caution when generalizing (private hospitals; STEEEP without ‘equity’).”
Response. We reinforced the Limitations: primarily applicable to Spanish private hospitals; extrapolation to public systems/other countries requires validation. We justify the omission of “equity” due to non‑measurability in our context.
Manuscript changes. Section 2.2 (STEEEP operationalization) and Limitations 4.3.
R1.3. “Undefined acronyms.”
Response. All acronyms are defined upon first mention (AI, KPI, EBITDA, SOM, STEEEP).
Manuscript changes. Abstract and Introduction.
Reviewer 2 Report
Comments and Suggestions for Authors
The manuscript can be of interest to an international audience due to the use of the Self-Organizing Maps (SOM) to analyze the selected indicators.
The manuscript discusses an interesting and actual issue. However, there are some issues that need to be clarified and improved.
- An explanation of what in concrete is SOM is needed in the methodological chapter. A methodology? How is this used in concrete
- How have the indicators been selected. Who decided which indicators were important?
- How has the study and the selection of indicators been validated?
- The authors write: “we standardized and consolidated operational and clinical information across the hospitals”. How? Who decided how they were standardized, based on what?
- The data collected is old. How do the authors work with accuracy of the data? What is in concrete a retrospective observational study? Which issues, situations have been observed and by whom? Please add references
- Many issues that in fact belong to the methodology chapter are presented as results. I suggest you consider the possibility to sample all issues related to the methodology chapter in the same chapter. It can helps the reader to follow and understand how data in fact has been sampled, analyzed, validated and classified.
- The authors write: “we identified the most effective KPIs 160 for healthcare organization management by integrating insights from existing literature….” Please add references. It is very hard to understand which indicators were selected, prioritized, analyzed etc.
- The authors wrote. “we adapted the STEEEP methodology for the private hospital management environment. This adaptation involved omitting the equity category, which includes indicators specific to the public sector and not applicable to private hospitals. “How was this work done in specific?
- Have the outcomes of the whole study been validated.
- Can this study be generalized and if yes to which audience?
- What is the specific contribution of this manuscript to management? and are the results valid for which type of organizations. Public. Private. Other?
- Any ethical consideration? Has patient information been used in the identification of the indicators?
- How consistent are the results of the study with the selected framework ?
- Have the authors of the manuscript participated in the analysis process?
Author Response
- Reviewer #2 (methodological traceability and design)
R2.1. “Clearly explain what SOM is and how it is used.”
Response. We added a specific SOM configuration subsection: 9×12 hex grid, Gaussian neighborhood, learning rate 0.05→0.01, 100 iterations, Euclidean distance, fixed seed; implemented in R 4.2.2 (kohonen/aweSOM). We also clarify that financial variables (Sales and EBITDA, current and with 3/6‑month lags) were excluded from training and subsequently projected onto the trained SOM.
Manuscript changes. Sections 2.6.1–2.6.3.
R2.2. “How were KPIs selected/validated; who decided?”
Response. We describe a multidisciplinary committee (clinical operations, patient experience, quality/safety, coding, finance), the STEEEP + Donabedian framework, and corporate availability (VMS).
Manuscript changes. Section 2.2 (framework and KPI governance).
R2.3. “Inter‑hospital standardization: how and according to which criteria?”
Response. We harmonized denominators, units, and operational definitions across hospitals and consolidated them daily in a corporate Data Warehouse with full traceability to source systems.
Manuscript changes. Section 2.3 (sources/standardization).
R2.4. “‘Old’ data; define retrospective observational design; add references.”
Response. 2019–2021 is the most recent homogeneous period before a corporate system change (2022). Design: retrospective, observational, multicenter; unit: hospital‑month; March–May 2020 excluded.
Manuscript changes. Sections 2.1 (design) and 4.3 (limitations).
R2.5. “Methodological aspects placed in Results.”
Response. We relocated to Methods the handling of missing data (≤10% temporal imputation), outliers (±3 SD by hospital–KPI), high‑correlation filter (|r|>0.90), and homogeneity analysis (Ward + Euclidean); Results now report the final composition only (47 KPIs; 14 hospitals; 21,714 observations).
Manuscript changes. Sections 2.4–2.6 (Methods) and section 3 (Results).
R2.6. “Validation of the study and selection of indicators.”
Response. We added robustness tests (seeds/parameters; resampling; k‑means) showing stable patterns and consistent topology/cluster structure.
Manuscript changes. Methods (Robustness/Validation) and Supplement (Fig. S4–S6; Table S2).
R2.7. “Ethical considerations; any patient data?”
Response. Only aggregated/anonymous hospital indicators were used; no patient‑level or identifiable data; GDPR compliant; IRB/consent not applicable.
Manuscript changes. Section 2.1 (ethics).
R2.8. “Generalization and target audience.”
Response. Primary audience: private hospitals; extrapolation to other settings is conditional on validation.
Manuscript changes. Discussion/Limitations.

Reviewer 3 Report
Comments and Suggestions for Authors
Major revision is recommended

Major revision is recommended
Author Response
- Reviewer #3 (theoretical framework / health economics)
R3.1. “Title does not indicate country/scope.”
Response. The title specifies a Spanish private multi‑hospital network.
Manuscript changes. Title page.
R3.2. “Weak abstract with undefined acronyms (EBITDA).”
Response. We rewrote the abstract, defined EBITDA, and clarified aims/methods/results/implications.
Manuscript changes. Abstract.
R3.3. “Introduction lacks context/motivation/gap/objectives.”
Response. We expanded the Spanish system context (dual coverage, sustainability), added motivation and the research gap (Spanish private sector under‑studied; non‑linear relations), and stated objectives with 0/3/6‑month horizons.
Manuscript changes. Introduction (opening paragraphs and closing aims).
R3.4. “Missing theoretical/conceptual framework.”
Response. We integrated Donabedian → IOM (six aims) → STEEEP and added Figure 1 (STEEEP → KPIs → SOM → finance).
Manuscript changes. Introduction and Figure 1.
R3.5. “Risk of multicollinearity; add basic correlations.”
Response. We applied a |r|>0.90 filter (with references) and provide the correlation matrix; Methods detail the rationale.
Manuscript changes. Sections 2.4–2.6 and Fig. S1; Results (section 3: final dataset composition and analysis).
R3.6. “Comparisons with the literature; robustness of findings.”
Response. We added comparative Table 2 and robustness tests (seeds, resampling, k‑means); we discuss lagged effects against prior evidence.
Manuscript changes. Discussion 4.3 and Methods (Robustness/Validation); Table 2.

Reviewer 4 Report
Comments and Suggestions for Authors
1. The authors should explicitly state how this study advances the literature beyond existing research on the relationship between hospital quality and financial performance, especially for using Self-Organizing Maps (SOM) for this purpose. The authors should provide a short comparative table with prior studies to strengthen this point.
2. The authors should consider a clear conceptual model linking STEEEP quality indicators, operational metrics, and financial outcomes in the introduction section.
3. The authors should provide more details on how the top five AI-based models were evaluated before selecting SOM, especially the criteria, scoring method, and any parameter settings for each model to enhance reproducibility.
4. Although the authors mentioned removing KPIs with high correlations, missing data, or outliers, the rationale for the correlation threshold (|r| > 0.90) and imputation choices should be supported with references or best practice guidelines.
5. The authors should describe in more detail the outlier correction method (e.g., replacement with median, winsorization) and how min–max scaling was applied in the presence of skewed data.
6. The discussion of the six KPI clusters could be enriched with practical managerial implications—how each cluster’s indicators can inform specific hospital management actions.
7. The authors stated that the rationale for choosing 3-month and 6-month lags is brief; I suggest the authors should consider supplementing with literature evidence or operational cycle logic to justify why these intervals are most relevant.
8. The authors should include potential biases from excluding two hospitals, handling missing data, and the generalizability of findings to other healthcare systems beyond the Vithas network.
9. Figures (e.g., SOM maps, dendrograms) would benefit from higher resolution, clearer legends, and annotations highlighting key findings. Adding a schematic workflow of the study process could improve reader comprehension.
10. Some sections contain redundant or overly long sentences. Tightening the writing, ensuring consistent terminology (e.g., STEEEP vs. STEEEPTM), and harmonizing KPI names across text, figures, and appendices would improve readability.
Author Response
- Reviewer #4 (clarity, parameters, managerial usefulness)
R4.1. “Explain how this study advances the literature; include a comparative table.”
Response. We made the contribution explicit (SOM on standardized STEEEP KPIs in a Spanish private network; 0/3/6‑month analysis) and added Table 2.
Manuscript changes. Introduction and Discussion 4.3; Table 2.
R4.2. “Clear conceptual model in the introduction.”
Response. Figure 1 shows the end‑to‑end conceptual flow.
Manuscript changes. Introduction (Figure 1).
R4.3. “Details on the evaluation of 5 AI models before selecting SOM.”
Response. We added a qualitative comparison (SVM, LDA, RF, k‑NN, SOM) with criteria (accuracy, scalability, interpretability, efficiency) and provided SOM parameters for reproducibility.
Manuscript changes. Section 2.6.1 and Fig. S3.
R4.4. “Justify correlation threshold, imputation, outliers, and normalization.”
Response. We detail min‑max scaling for all KPIs; outliers at ±3 SD per hospital–KPI with references in health‑services practice; ≤10% temporal imputation with methodological rationale; and the ex‑post projection of the dependent financial variables onto the trained SOM.
Manuscript changes. Sections 2.4–2.6 (data quality and analytical approach).
R4.5. “Justify 3‑ and 6‑month lags.”
Response. We motivate the 0/3/6‑month horizons using operational/financial cycles and prior evidence of lagged effects; we retain these horizons throughout the analysis.
Manuscript changes. Section 2.6 (analytical approach) and 4.2–4.3 (discussion).
R4.6. “Managerial implications by cluster.”
Response. We enriched the Discussion with actions linked to the six SOM clusters (e.g., A: scheduling/occupancy; B: inpatient process/documentation; C: surgical quality/experience; D: urgent activity; E: accessibility/continuity; F: evaluation/consents).
Manuscript changes. Section 4.2 (implications) and 4.1–4.3.
R4.7. “Improve figures/captions; terminology consistency (STEEEP).”
Response. We increased resolution, clarified legends/annotations, and harmonized terminology and KPI names across text, figures, and supplement.
Manuscript changes. Figures 2–3 and Supplement; style review.
